# CD26/DPP-4 in Chronic Myeloid Leukemia

**DOI:** 10.3390/cancers14040891

**Published:** 2022-02-11

**Authors:** Anna Sicuranza, Donatella Raspadori, Monica Bocchia

**Affiliations:** 1Hematology Unit, Department of Medical Science, Surgery and Neuroscience, University of Siena, 53100 Siena, Italy; bocchia@unisi.it; 2Hematology Unit, Azienda Ospedaliera Universitaria Senese, 53100 Siena, Italy; raspadori@unisi.it

**Keywords:** CD26, LSCs, CML, BCR-ABL, TKIs, TFR

## Abstract

**Simple Summary:**

CD26/dipeptidylpeptidase IV (DPP-4) is a membrane-bound multifunctional protein expressed in many primary solid tumors and in hematological diseases. Recent investigations demonstrated its specific expression on leukemic stem cells (LSCs) of Chronic Myeloid Leukemia (CML) bone marrow (BM) and peripheral blood (PB) samples. Thanks to this evidence, CD26 has been considered a novel exclusive stem cell marker of CML. The aim of this review is to describe and analyze the role of CD26 in the context of hematological malignancies and specifically of CML and to highlight its potential clinical application for the management of this disease.

**Abstract:**

CD26 expression is altered in many solid tumors and hematological malignancies. Recently, it has been demonstrated that it is a specific marker expressed on LSCs of CML, both in BM and PB samples, and absent on CD34+/CD38− stem cells in normal subjects or on LSCs of other myeloid neoplasms. CD26+ LSCs have been detected by flow-cytometry assays in all PB samples of Chronic-Phase CML patients evaluated at diagnosis. Additionally, it has been demonstrated that most CML patients undergoing Tyrosine Kinase Inhibitors (TKIs) treatment still harbored circulating measurable residual CD26+ LSCs, even when displaying a consistent deep molecular response without any significant association among the amounts of BCR-ABL transcript and CD26+ LSCs. Preliminary data of our Italian prospective multicenter study showed that CML patients with a poorer response presented with a higher number of CD26+ LSCs at diagnosis. These data confirmed that CD26 is a specific marker of CML and suggest that it could be considered for the monitoring of therapeutic responses.

## 1. Introduction

CD26/Dipeptidylpeptidase IV (DPP-4) is a 110 KDa type II glycoprotein. It is expressed both in soluble form in body fluids and as membrane-bound on the cell surface of several cells and tissues, particularly in the kidney, small intestine, endothelial, epithelial cells, and immune system cells [1]. CD26 is a multifunctional protein involved in numerous mechanisms, such as apoptosis, interaction with other proteins, hematopoiesis, and cancer progression [2,3]. The CD26 protein structure consists of two subunits containing three domains: a catalytic region, a transmembrane region, and a large extracellular region (Figure 1). The activity of CD26 can occur through two different manners: an enzymatic-dependent way or an enzymatic-independent action. In the first condition, CD26 digests and inactivates polypeptides, chemokines, and peptide hormones, such as the glucagon-like peptide or glucose-dependent insulinotropic peptide, whereas, for non-enzymatic action, it interacts with adenosine deaminase (ADA), caveolin-1 and other extracellular matrix (ECM) proteins.

Recent investigations demonstrated that CD26, when is in the soluble form (sDPP-4), induces cell signaling and lymphocytes proliferation [4], and that in human adipocytes, tumor necrosis factor-α (TNF-α) increases its release [5]. Moreover, data from studies focused on the niche of bone marrow (BM) of hematological diseases documented that an aberrant expression of CD26 could be involved in the persistence of leukemia stem cells (LSCs) [6,7].

## 2. Materials and Methods

We performed an electronic search to find papers in the PubMed database regarding CD26 and cancer or CD26 and hematologic malignancies. The search strategy included terms such as “CD26 and Cancer Stem Cells” or “DPP-4”, or “DPPIV” in combination with “hematological disease”. We also searched in the reference list of selected articles to perform more comprehensive research. In this review, we summarize all information extracted from original studies with the intent to offer a complete overview of the role of this interesting marker in the context of cancer and specifically of hematological malignancies. In particular, the review will also focus on the role of CD26 as a specific marker of CML LSCs and its potential clinical application for CML management and treatment.

## 3. CD26 and Cancer

Cancer stem cells (CSCs) are considered a tumor subpopulation with specific characteristics, such as tumor maintenance, resistance to chemotherapy, and ability of self-renewal and malignant transformation [8,9,10,11]. For many years, several studies investigated the role of CD26 on CSCs, reporting its involvement in gastrointestinal carcinoma, lung cancer, melanoma, and mesothelioma, resulting in its responsibility for malignant cell transformation [9,10,11]. However, given the great heterogeneity of different types of tumors and their microenvironments, and the variety of evidence that emerged, it is not possible to establish a unique and specific function for it. Thus, CD26 can be considered as a multifunctional protein involved in a series of processes related to its proteolytic activity, such as: (1) migration, (2) invasion, (3) immune modulation, (4) adhesion, (5) and involvement in apoptosis [12]. The non-enzymatic activity is influenced by the cell type and tumor microenvironment. Specifically, tumor progression could be favored by the interaction between CD26 and ECM proteins such as β catenin and E-cadherin. Moreover, it acts as a binding protein interacting with proteins such as fibronectin and collagen, resulting in association with adhesion migration and metastasis. About enzymatic activity, it contributes to the regulation of the SDF-1/CXCR4 axis, in which it exerts an antitumor action.

According to all these functions, CD26 can be considered as a tumor suppressor or a tumor promoter [10,11].

Here, we summarized several interesting data acquired from original studies regarding different functions related to CD26 expression in different types of cancer (Table 1).

Based on the collected data, we documented a controversial/opposite role of CD26, mostly related to its high or low expression in different types of cancer.

Studies on melanoma and non-small lung cancer (NSCLC) showed that the overexpression of CD26 acts as a tumor suppressor. This condition has been demonstrated by a significant reduction in invasiveness in melanoma cell lines and regulation of growth and tumorigenicity in NSCLC cells and mouse models [13,14,15].

Moreover, the CD26 expression on NSCLC cells was associated with the expression of several antigens involved in the tumor suppression (such as the fibroblast-activating protein or the CD44 antigen), confirming its role as a tumor suppressor in this malignancy [14].

However, clarifying a controversial aspect about the potential role of CD26 as an oncogene is needed. In recent work, it has been demonstrated that the CD26 expression in several NSCLC cell lines is associated with their invasive capacity and that treatment with a CD26 inhibitor, such as the flavonoid apigenin, suppresses CD26 expression and consequently inhibits tumor progression [16]. Based on this evidence, CD26 seems to have a double function, both as a tumor suppressor and as an oncogene. Therefore, further investigations are needed to better explain its specific role in NSCLC progression.

A similar role of CD26 has been demonstrated in colon CSCs, where the CD26 depletion is responsible for malignant behavior [17,18,19,20]. On the contrary, overexpression of CD26 in malignant mesothelioma (MPM) cell lines appears to increase metastatic diffusion [20]. Indeed, studies on MPM cell lines showed that the invasiveness role of CD26 is related to the expression of other surface markers such as CD9 and CD24 that, together with it, increase tumor formation in vitro and in vivo [21]. Moreover, the expression of CD26 upregulates periostin secretion by MPM cells, leading to enhanced MPM cell migratory and invasive activity [22]. Regarding breast cancer, CD26’s expression and relative role were found to be heterogeneous [23,24,25]. Results from in vitro experiments showed differences in CD26 expression that ranged between about 70% and 3% among the different histological subtypes [24].

Over the years, the role and the expression of CD26 have also been investigated in different hematological malignancies. Physiologically, CD26 is expressed on T cells. It has been reported that CD26 discriminates three subgroups of human CD4+ T cells with different responses to tumors: one with regulatory properties, one with a memory phenotype, and the third with a long-lasting stem memory characteristic. In particular, it has been demonstrated that human T cells with a high expression of CD26 on their cell surface produce a great quantity of cytokine IL-17, suggesting that the CD26 expression on CD4+ T cells could contribute to tumor regression [26].

In Chronic Lymphocytic Leukemia (CLL), CD26 expression, studied by using both flow cytometry and microarray gene expression, documented its overexpression in B-CLL [27,28].

A study comparing different B-cell lymphoid tumors proved that CD26 expression was variable. In follicular and mantle cell lymphomas, the CD26 was absent, whereas, in multiple myelomas and hairy cell leukemia, it was overexpressed. In addition, it was observed that the co-expression of CD26 with CD38 and ZAP70 was associated with a shorter time to treatment (TTT) [27]. Another study documented that the CD26 expression on B-CLL is associated with Rai’s stage, white blood cell count, β2-microglobulin, and lactic acid dehydrogenase (LDH) levels and showed that their expression impacts TTT. All these data confirmed the hypothesis that CD26 should be considered as a prognostic factor for the B-CLL patients’ risk assessment [29].

Studies conducted by flow cytometry tests on CD4+ T cells population identified a specific phenotype (CD26− or CD7− T cells with bright or dim CD3 or CD4 or bright CD7) that represents the phenotype for diagnosis of Sézary syndrome (SS) and mycosis fungoides and correlates with disease progression [30,31]. The loss of CD26 from the cell surface is a property for circulating SS cells. SS cells express CXCR4 whose ligand, SDF-1 is abundantly produced in the skin and normally inactivated by proteolytic scission by CD26. Experiments on CD26+ cell lines demonstrated that the SS cell’s chemiotaxis is influenced by soluble CD26 (sCD26): the presence or absence of sCD26 induces or inhibits the migratory response. These findings suggest that CD26 represents an important regulator of the SDF-1–CXCR4 axis by contributing to the SS cells recruitment in these patients [32].

An interesting study conducted on T-large granular lymphocyte (T-LGL) patients, investigated if the CD26 expression was associated with the clinical course. Data from experiments displayed that CD26+ patients had a worse disease requiring the use of hematopoietic growth factors and immunosuppressive treatment compared to that CD26− [33].

Regarding Multiple Myeloma (MM), little is known about the role of CD26. Recently, the CD26 expression was identified on human osteoclasts (OCs) and on plasma cells of MM patients. Moreover, it was observed that the CD26 expression resulted low or absent on MM cell lines cultured alone whereas intensively expressed on MM cell lines cultured in combination with OCs. Thus, it has been considered the possibility to employ the humanized IgG1 monoclonal antibody targeting CD26 (huCD26mAb) to inhibit the human OC differentiation and to reduce the MM tumor burden. These results suggest that CD26 is a potential target molecule in MM and that huCD26mAb could act as a therapeutic agent [34].

Acute leukemias have been studied to investigate phenotypes that could contribute to developing specific treatment strategies. An early study on a different type of acute leukemias (myeloid and lymphoblastic) investigated CD26 by evaluating its cell surface expression and its presence/absence as a soluble form. Data were compared with other hematologic diseases and showed that, although most acute leukemia patients displayed higher CD26 plasma levels than healthy controls, no clear correlation between plasmatic activity and surface expression was found [35,36]. However, even if the authors suggested that increased plasma levels of CD26 could represent a target for treating acute leukemias, given the discrepancies between studies, this issue remains to be clarified.

More recently, in a study evaluating the CD26 expression in both the compartments CD34+/CD38− and CD34+/CD38+ LSCs of several Ph+ and Ph− ALL samples, authors reported that only on ALL Ph+ p210 the CD26 was expressed [37]. Due to the expression being in the fraction CD34+/CD38− LSCs, with a phenotype very similar to that of chronic myeloid leukemia (CML), the authors further confirmed the relationship between the two entities [37].

Table 2 below summarizes the principal clinical and biological studies that investigated the role of CD26 in different hematological malignancies, as described above.

## 4. CD26 in Chronic Myeloid Leukemia: Characteristics of the Disease and Identification of CD26+ LSCs

Chronic myeloid leukemia (CML) is a myeloproliferative disorder characterized by a translocation between the breakpoint cluster region (BCR) on chromosome 22 and the Abelson gene (ABL) on chromosome 9.

The product of this cytogenetic alteration is p210 Bcr-Abl fusion protein with tyrosine kinase activity targeted by specific drugs, the so-called Bcr-Abl tyrosine kinase inhibitors (TKIs). CML history has dramatically changed in the last decade. Imatinib [38] and then second-generation TKIs (SG-TKI) nilotinib and dasatinib [39,40,41,42] became the most successful class of targeted therapies, exceeding all projected survival expectations. The exceptional success of TKI therapy in controlling disease for many years induced clinicians to discontinue TKI treatment with the intent to reach the ambitious condition of a “treatment-free remission” (TFR). However, despite achieving a deep molecular response (DMR), several CML patients lose the response after stopping TKI and require retreatment [43,44]. Available data suggest that this condition is due to residual circulating LSCs, probably resistant to TKIs [45].

In recent years, several in vitro and in vivo studies with the intent to characterize LSCs have been conducted. According to some authors, the presence of these residual CML LSCs is independent of BCR-ABL activity [46,47].

LSCs have been identified in the fraction CD34+/CD38−/Lin−. Valent and other authors described that CD34+/CD38−/Lin− CML LSCs specifically co-express CD26 and that its expression appears to discriminate CML LSCs from normal HSCs [6,48]. Thus, CD26 resulted in being more specific than other previously identified stem cell markers, such as IL1-RAP, CD25, and CD90, and emerged as a potential biomarker for the characterization and quantification of CML LSCs. Since 2014, the first studies conducted on BM samples from CML patients confirmed the presence of CML-specific CD34+ CD38− CD26+ LSCs, while CD26+ LSCs were not detected in BM of normal subjects and BM of patients affected by other hematological diseases. The authors showed that sorted CD26+ LSCs were confirmed to be BCR-ABL+ in short- and long-term colony cells derived from CD26+ and CD26− LSCs from CML patients. This condition was also confirmed in an NSG mouse model [6]. Culen et al., in a study on a cohort of 31 unselected CML patients, demonstrated that the percentage of CML CD26+ LSCs was correlated with leukocyte count [7]. In addition, a study performed by using a confocal laser microscope demonstrated a co-expression of Polycomb BMI1 protein and CD26+ antigen on CD34+/CD38− LSCs of BM samples of CML at diagnosis and during treatment with imatinib [49]. Normally, BMI1 is highly expressed in CML despite the reduction in the BCR-ABL1 transcript, exerting a negative impact on the outcome. However, it is still unknown if the expression of BMI1 in CML is in progenitors or mature cells. This study, for the first time, answered this question and, in addition to confirming that CD26+ represents a specific marker of stemness, demonstrated that the fraction CD34+/CD38−/CD26+ is the reservoir of the BMI1 protein [49].

Recently, a new study defining specific expression profiles in LSCs CD34+/CD38+ and CD34+/CD38− on AML and CML BM samples with the intent to identify novel targets has been performed [50]. The study was conducted by using multicolor flow cytometry, gene array, and quantitative polymerase chain reaction (qPCR) methods. Data documented that CML LSCs CD34+/CD38− exhibited the expression of a profile CD25+/CD26+/CD56+/CD93+/IL-1RAP+ in contrast with the AML LSCs profile only characterized by CD25+/CD96+/IL-1RAP+ and CD371+, without the expression of CD26, with the exception of the subgroup of AML FLT3-ITD+. Other surface markers, such as CD33, CD44, CD47, CD52, CD105, CD114, CD117, CD133, CD135, and CD184, were identified both in AML and CML LSCs but were also expressed on normal BM stem cells. In addition, the authors tried to evaluate a possible correlation between the expression of these surface markers in AML and CML LSCs and the Overall Survival (OS). Regarding CD26 expression, it has been observed to be correlated with a lowered OS in AML, but only in the subgroup FLT3-ITD+, which was the category expressing the antigen CD26 [50].

## 5. Role of CD26+ LSCs at Diagnosis of CML

Based on this evidence, also considering that the monitoring of residual LSCs from BM samples is not practical, our group first investigated the presence of CD26+ LSCs in peripheral blood (PB) samples compared to BM samples of Chronic-Phase (CP) CML patients at diagnosis using a flow cytometry assay [51]. We detected CD26+ LSCs in all CML samples at diagnosis. Figure 2 shows data comparable between PB and BM samples.

To estimate the number of CD26+ cells for microliter (cells/µL), we applied the following formula: [number of Whole Blood Counts (WBCs)/µL × (% of cells detected in the fraction CD45+/CD34+/CD38−/CD26+)] [51]. These data were confirmed in another study in which we validated our flow-cytometry method in a larger series and documented the presence of CD26+ LSCs in PB samples of all 211 newly diagnosed CML patients tested. [52]. Given these confirmatory results, the flow cytometry measurement of a CD34+/CD38−/CD26+ population in PB using custom-made, lyophilized, pre-titrated antibody mixture tubes resulted in an easy, fast, and standardized method useful for the diagnosis of CML.

## 6. Role of CD26+ LSCs for Minimal Residual Disease and Treatment-Free Remission

In addition to CML patients at diagnosis, we also studied PB samples of CML patients during first-line TKI treatment (imatinib, nilotinib, dasatinib) and during TFR. From a total of 236 PB samples measured, circulating CD26+ LSCs were detectable in 169/236 (71.6%) of patients [51]. No correlation between molecular response and the number of residual LSCs was found, reinforcing the hypothesis that LSCs may be “invisible” to a standard quantitative real-time PCR (qRT-PCR) BCR-ABL assay. Our study demonstrated that most CML patients still harbor circulating residual LSCs and suggest that molecular response mainly refers to the disappearance of proliferating CML precursors while quiescent, TKI-resistant LSCs are probably less detectable by qRT-PCR. Another important proof that emerged from our investigations is that residual PB CD26+ LSCs were also found in CML patients during a stable and prolonged TFR. However, an inverse correlation between the number of circulating CD26+ LSCs and the duration of TFR was found [51]. Based on these intriguing data, we started a prospective multicenter Italian study (“Prospective flow cytometry evaluation of circulating residual leukemia stem cells in CML patients during TKIs treatment”—Prospective FLOWER Study) to monitor PB CD26+ LSCs at diagnosis and during TKI therapy at specific time points (+3, +6, +12, and +24 months of treatment) and after discontinuation, which is currently ongoing. Interim results confirm the persistence of residual CML-specific CD34+/CD38−/CD26+ LSCs in CML patients under TKIs treatment, even in DMR. Of note, the number of circulating CD26+ LSCs does not correlate with the number of BCR-ABL1 copies [53].

A recent work of a Turkish group on 38 CML patients showed that CD26+ LSCs were fluctuating but measurable, both in PB and BM samples, during TKI treatment. However, in those CML patients in DMR, CD26+ LSCs were present in BM but not in PB samples [54]. These findings, even if based on a small cohort of patients evaluated, once again open the controversial question about the appropriateness of PB and/or BM samples for measuring residual CD26+ LSCs. Further studies and new data are required to better understand this condition, considering the importance of this information for the choice of a possible TKI discontinuation.

## 7. CD26+ LSCs and Immune System

Given the persistence of CD26+ LCSs in a subset of CML patients in stable and prolonged TFR, the question arises regarding the interaction of CML LSCs and the immune system. As such, we decided to explore the expression on CD26+ LSCs of specific markers able to modulate the immune system, such as PD-L1.

Recently, immune escape mechanisms in CML have been identified to directly hamper cellular immunity and restored immune T cells effectors and decreased PD-1 and immune suppressor T cells have been shown in patients achieving DMR and sustained TFR [55,56,57]. Our novel flow cytometry approach to identify circulating CML LSCs would offer a unique possibility to further explore the relationship between the immune system and resistant CML reservoir. Preliminary data showed that the expression of PD-L1 on CML CD26+ LSCs is different between patients, and only about half of them show PD-L1+ CD26+ LSCs [53]. Therefore, at diagnosis, a higher CD26+ LSCs number, PD-L1 positivity, or both may correlate with a lower probability of achieving an optimal response. However, enrolment and follow-up are ongoing, and additional studies are needed to confirm this observation.

## 8. CD26+ LSCs, Novel Approach, and Therapeutic Role

Given the evidence of TKIs resistance and relapse after therapy related to the presence of quiescent residual LSCs, several recent studies explored and proposed new potential strategies to target and consequently eradicate CD26+ cells. In 2014, some authors tried to explore this hypothesis by incubating CML LSC with vildagliptin before injecting these cells into NSG mice. Gliptin exposure resulted in reduced engraftment of CML cells. Afterward, they examined CML patients treated with nilotinib receiving a gliptin and observed that after gliptin treatment, BCR-ABL1 levels substantially decreased to the point of being undetectable [6].

Recent work proposed an innovative method to selectively target CML CD26+ LSCs by using an anti-CD26 (Belogemab) immunoliposome loaded with venetoclax (IL-VX). In vitro experiments demonstrated that after two days of treatment, free venetoclax did not induce apoptosis in the CMLT1 cells, even at high concentrations, while the treatment with IL-VX reduced CD26+ LSCs. In addition, the study showed a strong synergism between IL-VX and TKIs, as documented by the treatment of CMLT1 cell lines with different concentrations of IL-VX, nilotinib, and imatinib. These interesting results confirmed the selectivity of IL-VX in CML patients, offering the possibility to eradicate residual CD26+ LSCs and establish the drug concentration necessary to induce apoptosis [58]. On the contrary, another in vivo study evaluating the antileukemic effect on growth and survival of CML LSCs of the DPP-4 blocker vildagliptin combined with imatinib or nilotinib in an NSG mouse model did not show a significant synergic effect. Results of experiments showed that when the CD34+/CD38− CML LSCs have been incubated with nilotinib, they were induced to increase their spontaneous apoptosis ability. On the contrary, incubation with a combination of nilotinib and vildagliptin did not increase apoptosis compared to nilotinib alone [59].

Albeit these investigations are very interesting and could open the way to new therapeutic strategies, considering these contrasting observations, additional studies are needed.

Recent data proved that some miRNA, such as miR-300 or miR-126-3p, are involved in the regulation of the activity of CML LSCs. Specifically, miR-300 induces an antiproliferative function [60], while miR-126-3p controls the self-renewal of LSCs [61]. Based on this evidence associated with miRNAs, an Argentinean study proposed a new pharmacogenomic tool to better identify CML LSCs by exploring, with next-generation sequencing (NGS) technology, the profile of the miRNoma of CD26+ LSCs and HSCs. The study revealed that in both CML LSCs CD26+ and CD26− several miRNAs were down-regulated; only the expression of miR-196a-5p was documented in LSCs CD26+ at diagnosis with levels nine-fold more than LSCs CD26−. Additionally, miRNAs were involved in the metabolism of LSCs [62].

Together with the other evidence reported, these findings offer interesting strategies for identifying new therapeutic approaches for the treatment of CML patients.

## 9. Conclusions

Over the years, a large number of investigations on CD26 focused attention on this marker, offering a better understanding of its contribution to cancer. Even if its functional role remains complex and needs more investigations, the data obtained from several studies and our recent experience about its expression on CML LSCs demonstrate the importance of its characterization. CD26 is a marker allowing us to follow CML in a different way (such as by evaluating the cellular response), with the opportunity to understand other aspects of its biology and interactions with the immune system.

In addition, all these encouraging results and future studies could demonstrate the role of CD26 as an important target for developing a therapy aimed at eradicating CML.

## Figures and Tables

**Figure 1 cancers-14-00891-f001:**
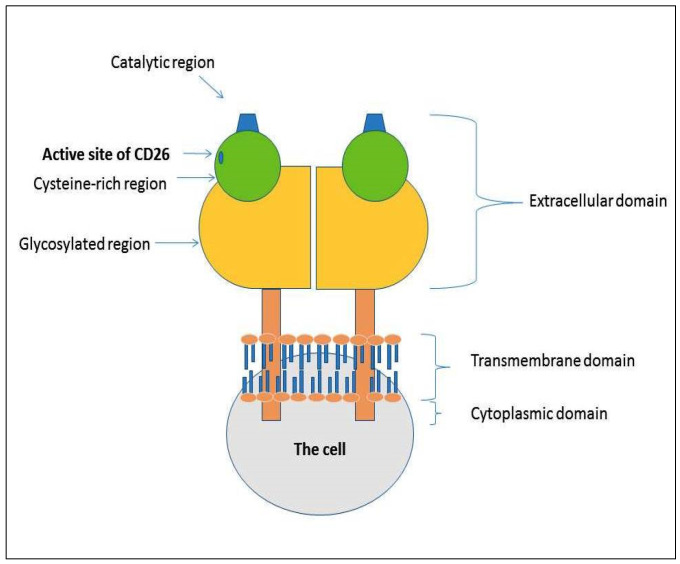
Structure of CD26. It consists of two subunits containing three domains: catalytic region, a transmembrane region, and a large extracellular region.

**Figure 2 cancers-14-00891-f002:**
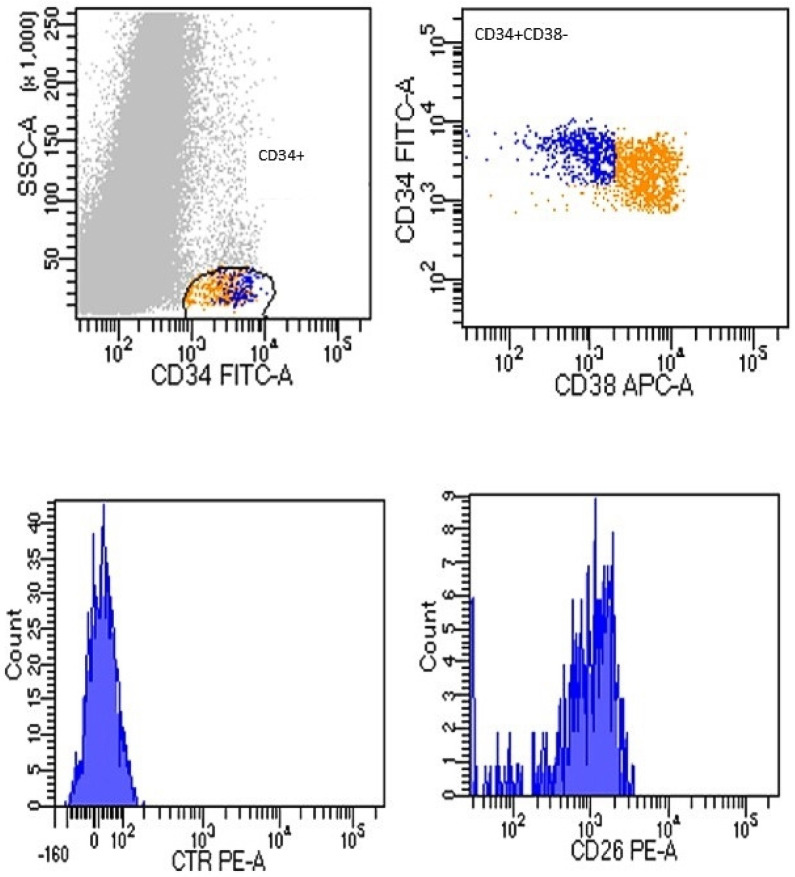
Flow cytometry CD26 expression on PB sample of CML at diagnosis. After a selection of cellular compartment CD34+/CD38− (**upper** boxes), the fraction CD26+ was selected within them (boxes on **down**, represented as histograms). Acquisition of at least 1 × 10^6^ cells was performed by using FACSCanto II flow cytometry instrument (BD, Biosciences, San Jose, CA, USA).

**Table 1 cancers-14-00891-t001:** Summary of clinical and biological studies that investigated the role of CD26 expression in different types of cancer.

Reference	Disease	CD26 Expression	Effect of CD26
[13]	Melanoma	Overexpression	Tumor suppressor
[14,15,16]	NSCLC	Overexpression	Tumor suppressor/ Tumor promoter
[17,18,19,20,21,22]	Colon cancer MPM	Reduction Overexpression	Tumor promoter
[23,24,25]	Breast cancer	Heterogeneous expression	Tumor suppressor

Abbreviations: NSCLC, non-small cell lung cancer; MPM, malignant pleural mesothelioma.

**Table 2 cancers-14-00891-t002:** Summary of clinical and biological studies that investigated the role of CD26 (expressed as soluble form or as surface glycoprotein) in different hematological malignancies.

Reference	Disease	CD26 Presence/Absence	Effect of CD26
[27,28,29]	Follicular/mantle lymphomas	CD26− membrane expression	-
	B-CLL	CD26+ B-cell membrane expression	Disease progression
[30,31,32]	SS	Presence of soluble CD26	Induction of chemiotaxis
		Absence of soluble CD26	Inhibition of chemiotaxis
[33]	T-LGL LPD	CD26+ T cells	Disease progression
		CD26− T cells	Moderate clinical behavior
[34]	MM	CD26+ MM cells	Increase of MM tumor burden
[35,36]	AML	High CD26 plasma levels	Disease progression
[37]	ALL	CD26+ Ph+ p210	Disease progression

Abbreviations: B-CLL, B-Chronic Lymphocytic Leukemia; SS, Sezary syndrome; T-LGL LPD, T-large granular lymphocyte lymphoproliferative disorder; MM, Multiple Myeloma; AML, Acute Myeloid Leukemia; ALL, Acute Lymphoblastic Leukemia; Ph+, Philadelphia Positive; OS, Overall Survival.

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
