# Peer review of "CD26/DPP-4 in Chronic Myeloid Leukemia"

_cancers, 2022, doi:10.3390/cancers14040891_

Round 1

Reviewer 1 Report

This work is a Review article regarding CD26 and its role in chronic myeloid leukemia. The Review portion of this article is a straightforward summary. However, there are instances throughout this Review where the authors refer to new data that has not been peer reviewed (‘interim unpublished results’). This should be removed from the article and the article should be resubmitted before a full review can be provided.  

Reviewer 2 Report

With this review the authors explore the potential of CD26 as a specific marker for CML.

General comments

Overall is not clear what is the key contribution to the field brought by this review. Information here presented can be found in other reviews (e.g. https://www.nature.com/articles/s41375-019-0490-0; https://www.ncbi.nlm.nih.gov/pmc/articles/PMC7990348/).

The authors should use the same definition throughout the text and not have “CD26/DPP-4”, only “CD26” or only “DPP-4”. Is not clear for the reader.

The writing must be improved. Sentences are not clear. Some examples are given in the “Specific comments below”. Also the references should be placed correctly after the sentence to which they refer to.

The text and Figure 1 should be better coordinated. The terms in text do not match the ones in the figure. A more descriptive figure legend should be included.

Figure legends should be improved. Figures that summarize better the text should be included.

In the topic “CD26/DPP-4 and cancer”, it is referred that CD26 can have both tumor suppressor or oncogene function. But for example for NSCLC in the literature one is able to find both examples as a tumor suppressor or oncogene. In addition, why is not included in this summarizing table (Table 1) the hematological malignancies discussed below?

For all the topics on “4. CD26/DPP-4 in hematological malignancies” only one or few examples are given and do not allow to have the complete idea

Is the data on Figure 2 the same as published before (ref 51)? This does not add much information to the text.

Specific comments

Lines 18-19: The sentence “CD26+ LSC resulted measurable by flow-cytometry in 18 100% of PB samples of Chronic Phase CML patients at diagnosis.” is not clear and must be revised

Lines 31-33: The sentence “ It is expressed both in soluble form in body fluids and as cell surface glycoprotein in several cells and tissues, particularly in the kidney, small intestine, endothelial, epithelial cells and also on immune system cells” must be revised. CD26 is a glycoprotein regardless of being in the soluble form or as a transmembrane protein.

Line 40: not clear to what “independent action.“ refers to. Is it enzymatic independent action?

Line 41-42: for the enzymatic independent is not clear what is the “action” of CD26, the only comment is “for non-enzymatic action it interacts with ADA, calveolin-1 and others extracellular matrix (ECM) proteins.” and here is not clear the action

Lines 48-51: sentence is not clear and must be rewritten.

Line 65: revise “From many years, “ should it be “For many years”?

Reviewer 3 Report

In this review article, the authors described CML-specific expression of CD26/DPP-4 in the stem cell fraction in contrast to other hematological malignancies and solod cancers. They discussed the clinical implication and future perspectives of CD26+LSC in MRD-detection and therapy of CML.

Although it is well written in general, the following points need to be reconsidered.

  • Abstract and text (Line 284-287) state that the number of CD26 + LSC in peripheral blood does not correlate with the number of BCR-ABL1 copies, even though it is ongoing preliminary data. This is an important point regarding whether the CD26 + LSC number is useful for MRD-detection, and seems to be inconsistent with the paper by Herrmann et al. (Blood 2014). It might confuse the readers and would be avoided, or detailed data should be fully presented and discussed.
  • Is Table 1 necessary? Rather, the table showing that CD26 is specific to the LSC of CML not of other leukemia would be more important.
  • It is also desirable to add consideration to the biological mechanism and significance of CD26 expression.

Round 2

Reviewer 3 Report

The revised version is acceptable.

Author Response

Dear Reviewer, 

I send you the final version of  the manuscript after my revisions and after the English Editing service.

I hope it is ok for you.

Best regards,

Anna Sicuranza
